# Poor Eating Behaviors Related to the Progression of Prediabetes in a Japanese Population: An Open Cohort Study

**DOI:** 10.3390/ijerph191911864

**Published:** 2022-09-20

**Authors:** Yuichiro Otsuka, Tomoko Nakagami

**Affiliations:** 1Division of Public Health, Department of Social Medicine, Nihon University School of Medicine, Tokyo 173-8610, Japan; 2Division of Diabetology and Metabolism, Department of Internal Medicine, Tokyo Women's Medical University School of Medicine, Tokyo 162-8666, Japan

**Keywords:** prediabetes, normoglycemia, progression, remission, eating behavior, body mass index

## Abstract

This study aimed to examine lifestyle factors associated with the change in glucose categories among individuals without diabetes. We analyzed cohort data of medical check-ups at baseline between April 2008 and December 2012. The primary and secondary outcomes were the change in glucose categories from normoglycemia (glycated hemoglobin (HbA1c) < 5.7% and fasting plasma glucose (FPG) < 5.6 mmol/L) to prediabetes (HbA1c 5.7–6.4% or FPG 5.6–6.9 mmol/L) and from prediabetes to normoglycemia. During a mean follow-up of 2.4 years, 7083 of 57,018 individuals with normoglycemia developed prediabetes, whereas 4629 of 9926 individuals with prediabetes returned to normoglycemia. Factors associated with progression to prediabetes were baseline BMI (hazard ratio [95% confidence interval]: 1.08 [1.07–1.09]), change in BMI during follow-up (1.05 [1.03–1.07]), late dinner/snacking (1.16 [1.10–1.22]), skipping breakfast (1.12 [1.06–1.18]), and heavy alcohol consumption (1.33 [1.24–1.42]). Factors associated with return to normoglycemia from prediabetes were baseline BMI (0.94 [0.93–0.95]) and change in BMI during follow-up (0.95 [0.93–0.97]). In conclusion, poor eating behaviors, such as skipping breakfast, late dinner/snacking, and heavy alcohol consumption, were associated with the progression from normoglycemia to prediabetes.

## 1. Introduction

Prediabetes is an intermediate hyperglycemic state in which blood glucose levels are higher than normal but below the diabetic threshold [1]. The American Diabetes Association (ADA) defines prediabetes as impaired fasting glucose (IFG; i.e., fasting plasma glucose (FPG) 5.6–6.9 mmol/L), impaired glucose tolerance (IGT; i.e., 2 h plasma glucose of 7.8–11.0 mmol/L on 75 g oral glucose tolerance test (OGTT)), or an elevated glycated hemoglobin (HbA1c) of 5.7–6.4% (38–46 mmol/mol)) [2].

Prediabetes has been recognized as a high-risk state for future diabetes [1,3], early diabetic microvascular complications [4,5], cardiovascular events, and all-cause mortality [6]. Nonetheless, some individuals with prediabetes return to normoglycemia. In a population-based cohort study that surveyed the natural history of diabetes in England, 55% of individuals with FPG of 6.1−6.9 mmol/L and 80% of those with FPG of 5.6–6.0 mmol/L at baseline returned to normal FPG < 5.6 mmol/L after a 10-year follow-up [7]. However, a meta-analysis of 16 randomized controlled trials showed that lifestyle intervention reduced the progression from prediabetes to diabetes by 54% after a 1-year follow-up and by 36% after a 3-year follow-up [8]. Particularly, the most important determinant of risk reduction for incident diabetes in these trials was weight loss [8,9].

Some studies have reported unfavorable eating behaviors, including skipping breakfast [10], snacking [11], and fast eating speed [12], as being factors related to the progression from non-diabetes to diabetes, independent of body mass index (BMI). However, it is unclear whether the effects of eating behaviors contribute to the development of diabetes independent of BMI, as other studies have shown conflicting results [13,14]. Furthermore, only a few studies have evaluated the comprehensive effects of lifestyle behaviors on the deterioration from normoglycemia to prediabetes and the return to normoglycemia from prediabetes. Thus, this study aimed to examine the association between lifestyle factors, including eating behaviors, and the change from normoglycemia to prediabetes, as well as the change from prediabetes to normoglycemia, independent of baseline BMI, and the change in BMI during follow-up.

## 2. Materials and Methods

### 2.1. Survey Procedure, Design, and Participants

Figure 1 shows the participant selection flowchart. This was a retrospective, open cohort study performed on participants who underwent an annual health check-up at the medical examination center of a hospital in Tokyo, Japan. A baseline survey was conducted between April 2008 and December 2012. Participants aged 18–86 years at baseline were included, and those with diabetes at the baseline survey and with follow-up <1 year were excluded. Those with missing data were also excluded from the analyses. Consequently, among 112,304 individuals screened in the baseline survey, 66,944 participants who fulfilled the inclusion criteria were included in the analysis.

### 2.2. Glucose Tolerance Status

Diagnoses of diabetes at baseline and during follow-up were determined by a panel of physician investigators. Diabetes was defined by a self-reported use of glucose-lowering medication, FPG level of ≥7.0 mmol/L, or HbA1c level of ≥6.5%, according to the Japan Diabetes Society, ADA, and World Health Organization [3,15,16]. Prediabetes was defined as an FPG level of 5.6–6.9 mmol/L or an HbA1c level of 5.7–6.4%, based on the ADA criteria [3]. In the present study, diagnosis of diabetes or prediabetes on 2 h plasma glucose values on an OGTT was not applied because an OGTT was not included in the general health check regimen.

### 2.3. Lifestyle Behaviors

The exposure variables consisted of 10 self-reported question items regarding lifestyle behaviors created by the Ministry of Health, Labour and Welfare of Japan [17]. This questionnaire survey was administered at the same time as the medical examination. The items included three exercise behaviors (physical activity, regular exercise, and walking speed), four eating behaviors (eating speed, eating dinner within 2 h before bedtime for ≥3 per week, snacking after dinner for ≥3 per week, and skipping breakfast ≥ 3 per week), subjective sufficient sleep, habitual smoking, and frequency and amount of alcohol consumption. These variables were analyzed as categorical variables based on the response options. Non-regular exercise was defined as no exercise twice or more per week for ≥30 min over the past 1 year or more. Less physical activity was defined as no walking or performing an equivalent activity for >1 h per day. Fast walking speed was defined as walking faster than other same-age individuals. Eating behaviors were assessed using the following questions: How fast is your eating speed compared with others around the same age? (fast, normal, or slow), Do you have an evening meal within 2 h before bedtime ≥ three days per week? (yes or no), Do you eat a snack after your dinner ≥three days per week? (yes or no), and Do you skip breakfast ≥three days per week? (yes or no). Adequate sleep was defined as a subjective feeling of refreshed sleep after waking up. Habitual smoking was defined as the total number of cigarettes smoked >100 for over 6 months and smoking in the previous month. For alcohol consumption, an average daily consumption of >40 g was considered heavy alcohol consumption. Particularly, those who were (1) drinking “every day” and drinking more than two drinks per day and (2) those who drink “sometimes” and drink more than three drinks per day were classified as heavy drinkers. The ethanol content per drink was calculated as equivalent to 20 g.

### 2.4. Covariates

Body weight was measured with the patients in light indoor clothes, and BMI was calculated as weight divided by squared height (kg/m^2^). A change in BMI was defined as the value obtained by subtracting baseline BMI from BMI at the onset of the outcome or at the final follow-up. Hypertension was defined as systolic blood pressure ≥140 mmHg, diastolic blood pressure ≥90 mmHg, or self-reported use of antihypertensive medications, based on the Japanese Society of Hypertension guidelines [18]. Dyslipidemia was defined as high-density lipoprotein cholesterol levels <1.0 mmol/L, low-density lipoprotein cholesterol levels ≥3.6 mmol/L, triglyceride levels ≥1.7 mmol/L, or self-reported use of lipid-lowering medications based on the Japan Atherosclerosis Society guidelines [19]. All the above items and age, sex, and family history of diabetes were treated as covariates in the study.

### 2.5. Statistical Analysis

First, participants’ characteristics at baseline were compared between glucose statuses (i.e., normoglycemia and prediabetes) at baseline. Continuous variables are expressed as the median and the upper and lower quartiles, whereas categorical variables are presented as *n* (%). Continuous variables were compared using the Mann–Whitney test because the age and BMI data were not normally distributed, whereas categorical variables were compared using the chi-square test. Second, the change rates and their 95% confidence interval (CI) according to the two glucose tolerance statuses during follow-up were calculated. Third, Cox proportional hazards regression was used to estimate the hazard ratios (HRs) and their 95% CIs for normoglycemia and prediabetes. The exposure variables were physical activity, walking speed, regular exercise, eating speed, late dinner/snacking, skipping breakfast, insufficient sleep, smoking status, and heavy alcohol consumption. The covariates were age, sex, baseline BMI, change in BMI during follow-up, family history of diabetes, hypertension, and dyslipidemia. To determine these variables, we referred to the factors associated with glucose tolerance in previous studies [10,11,12,20,21,22]. The Schoenfeld residuals test and graphical methods were used to evaluate the Cox proportional hazard assumption. Fourth, we investigated the hierarchical effects of poor eating and exercise behaviors on the progression to prediabetes in a sensitivity analysis. Poor eating or exercise behaviors were determined by summing the number of three poor eating behaviors (fast eating speed, late dinner/snacking, and skipping breakfast) or three poor exercise behaviors (less physical activity, slow walking speed, and non-regular exercise), respectively. Cox proportional hazards regression was used to examine the effect of multiple poor eating and exercise behaviors on the change in glucose tolerance status.

All statistical analyses were performed using Stata version 17.0 (StataCorp, College Station, TX, USA). All reported *p*-values were two-tailed, and statistical significance was set at *p* < 0.05.

## 3. Results

### 3.1. Comparison of Baseline Characteristics in Individuals with Normoglycemia and Prediabetes

Table 1 shows the comparison of baseline characteristics by glycemic status. A total of 57,018 and 9926 participants were categorized into normoglycemia and prediabetes groups, respectively. The prediabetes group was significantly older, included more men, had a higher BMI, had higher rates of hypertension and hypertriglyceridemia, had less physical activity, was eating at faster speeds, included a higher number of smokers, and was more likely to be heavy alcohol consumers than the normoglycemia group. Additionally, the normoglycemia group was significantly more likely to engage in non-regular exercise, late dinner/snacking, skipping breakfast, and having insufficient sleep than the prediabetes group.

### 3.2. Change Rate of Glucose Tolerance and Mean Change in BMI during Follow-Up

The overall mean follow-up period was 2.6 years, with an average follow-up period of 2.7 years among those who had prediabetes at baseline and 2.6 years among those with normoglycemia at baseline. Overall, 7083 out of 57,018 individuals with normoglycemia at baseline (12.4%) progressed to prediabetes status during follow-up (change (incidence) rate: 50.4 per 1000 person-years (95% CI: 49.2–51.6)). On the other hand, 4629 out of 9926 individuals with prediabetes at baseline (46.6%) returned to normoglycemia status during follow-up (change (return to normoglycemia from prediabetes) rate: 232.2 per 1000 person-years (95% CI: 225.6–239.0)).

The mean change in BMI among individuals who developed and did not develop prediabetes from normoglycemia was 0.56 ± 1.36 kg/m^2^ and 0.26 ± 1.16 kg/m^2^ (*p* < 0.001), respectively. The corresponding values were −0.09 ± 1.34 kg/m^2^ and 0.17 ± 1.12 kg/m^2^ (*p* < 0.001) in individuals who returned and not returned to normoglycemia from prediabetes.

### 3.3. Association between Lifestyle Behaviors and the Change in Glucose Status during Follow-Up

Table 2 shows the HRs of baseline lifestyle factors for the progression to prediabetes from normal glucose at baseline and for the return to normoglycemia from prediabetes at baseline. Taking late dinner/snacking (HR: 1.16, 95% CI: 1.10–1.22), skipping breakfast (HR: 1.12, 95% CI: 1.06–1.18), and heavy alcohol consumption (HR: 1.33, 95% CI: 1.24–1.42) were associated with the progression to prediabetes from normoglycemia. In contrast, there were no significant lifestyle factors related to the return from prediabetes to normoglycemia. On the other hand, baseline BMI and the change in BMI were significantly associated with the change in glucose status. They showed a positive association with the progression to prediabetes from normoglycemia and a negative association with the return to normoglycemia from prediabetes.

### 3.4. Sensitivity Analysis of Poor Eating and Exercise Behaviors Associated with Progression to Prediabetes from Normoglycemia

Multivariate Cox proportional hazards models (Table 3) showed that having one, two, and three poor eating behaviors was linearly associated with progression to prediabetes (HR: 1.06, 95% CI: 1.00–1.13; HR: 1.14, 95% CI: 1.07–1.22; and HR: 1.21, 95% CI: 1.10–1.34, *p*-trend < 0.001). In contrast, poor exercise behaviors were not associated with progression to prediabetes.

## 4. Discussion

The overall effect of lifestyle behaviors on the improvement in glucose status, particularly in lean individuals, is yet to be clarified. This study revealed that lifestyle behaviors, including late dinner, skipping breakfast, and heavy alcohol consumption, were associated with the progression to prediabetes, independent of baseline and change in BMI. However, there were no lifestyle behaviors associated with the return from prediabetes to normoglycemia, independent of baseline and change in BMI. To our best knowledge, this study is the first to examine poor eating and drinking behaviors related to the progression of prediabetes in the Japanese population. These findings have important implications for the development of strategies to prevent prediabetes.

The incidence of prediabetes [23,24] and the return rate from prediabetes to normoglycemia [25] differ across various studies, including our study. This may be due to differences in the characteristics of the studied population, observational periods, and the applied definitions of prediabetes.

Several studies reported that skipping breakfast was associated with poor glycemic control in healthy individuals [26] and in those with incident type 2 diabetes [10,11,12] and was only partly mediated by BMI [27]. One randomized clinical trial showed that skipping breakfast led to lower levels of insulin and intact glucagon-like peptide-1 and higher levels of glucagon and fatty acids after lunch and dinner [28]. These hormonal movements increased the average glucose levels all day, as compared with those at breakfast consumption [28]. These findings support that skipping breakfast could increase blood glucose levels independent of BMI in the present study.

Previously, late-night dinner was associated with hyperglycemia even after adjusting for BMI in a cross-sectional data of 61,364 healthy Japanese [29]. In our cohort data, late dinner/snacking was associated with the progression to prediabetes, independent of baseline and change in BMI. This supports the findings of a previous report that late-night dinners may enhance postprandial blood glucose response and 4 h average blood glucose level [30].

A Japanese 3-year cohort study showed that fast eating is a strong risk factor for incident type 2 diabetes [12]. However, fast eating speed was not associated with progression to prediabetes, and the HR was significantly higher in people with normal eating speed than in those with the slow eating speed in the present study. The survey period, age range, and targeted glucose tolerance status may have influenced these associations. In general, eating fast leads to obesity by making it difficult for the brain to recognize calories, thus delaying satiety and resulting in excessive calorie absorption [31]. Thus, the magnitude of the eating speed on the progression to prediabetes may have been diluted after adjusting for BMI, as well as the change in BMI.

An increase in poor eating behaviors was linearly associated with a higher risk of prediabetes in the current study. In contrast to a previous meta-analysis that exercise intervention could improve glucose tolerance [32], poor exercise behaviors were not associated with the progression to prediabetes in the present study. There are several possible explanations for this finding. First, individuals with exercise behaviors, such as moderate-intensity physical activity, may have an intake of more calories than those with non-exercise and dilute the benefit of exercise behaviors. Second, the question of exercise behaviors in this study was too simplistic and subjective, and we could not calculate the metabolic equivalent of a task, thus limiting the power of the study to estimate the prediabetes condition. Third, the participants with exercise behaviors in this study could be a high-risk population for prediabetes. A 3-year cohort study of the general Japanese population that used the same questionnaire as in our study showed that habits of exercise and physical activity were positively associated with the incidence of diabetes [33].

Our study showed that smoking was not a risk factor for prediabetes in contrast to other studies [34]. The conflicting finding might be due to differences in the amount and duration of smoking [35]. Our study identified heavy alcohol intake as an independent factor for prediabetes, as shown in a previous report [36]. Heavy alcohol intake alters glucose metabolism via insulin resistance in the liver [37].

The present study showed that body weight and changes in body weight had different risks related to the progression to prediabetes and the return to normoglycemia from prediabetes. The independent impact of the three eating behaviors remained associated with the progression of prediabetes but disappeared during the return to normoglycemia. This supports the importance of comprehensive lifestyle modifications leading to weight loss for the transition from prediabetes to normoglycemia, as previously proven in lifestyle intervention trials (the Diabetes Prevention Program (DPP) and Finnish Diabetes Prevention Study (FDPS)) for prediabetes [38,39]. The reason for the independent association of the three eating behaviors (namely, late dinner/snacking, skipping breakfast, and heavy alcohol consumption) and the progression to prediabetes was not known. These eating behaviors may contribute to rapid glucose concentration and, subsequently, to a decrease in insulin secretion and β-cell dysfunction [28,40,41], which is a different contribution to the progression of prediabetes from insulin sensitivity related to BMI and its change.

This study has some limitations. First, this was a retrospective, single-center, open cohort study in Japan, thus creating possible selection bias. Individuals with severe illnesses tend to visit medical institutions and may not participate in annual physical check-ups. Moreover, as data from health check-ups were used, the participants may have been more health-conscious or healthier than the average population. Thus, the findings have weak external validity. Future studies should gather nationally representative samples. Second, misdiagnosis of prediabetes might have occurred in some cases because we relied on a single result of the FPG or HbA1c test. Indeed, the prevalence of diabetes, as defined by FPG, would decrease by approximately 24% if a confirmatory FPG test was performed 2 weeks later [42]. In the health check-up system, we lacked access to further information, such as continuous determination of HbA1c, glucose, and OGTT. In addition, our study did not include OGTT data. Thus, individuals with IGT results that were within the normal ranges of FPG and HbA1c, the target of lifestyle intervention studies (DPP, FDPS) [38,39], were not included as our participants with prediabetes. This might have influenced our results. Third, although we adjusted for potential confounding behaviors, there were no data on socioeconomic status, which is an important factor for diabetes [43]. Fourth, lifestyle behaviors were based on self-reports. Thus, the judgment was qualitative and might have a reporting bias. Nonetheless, the questionnaires were created by the Ministry of Health, Labour and Welfare of Japan based on existing evidence. Fifth, the obesity indicator was limited to BMI, and waist circumference as an indicator of visceral obesity was not included. Despite these limitations, the major strength of our study was the large sample size with a broad age range and cohort data.

## 5. Conclusions

Poor eating and heavy alcohol drinking behaviors were associated with the progression to prediabetes from normoglycemia, independent of baseline BMI and change in BMI. For future research, more effective lifestyle interventions according to individual behaviors are needed to prevent impaired glucose tolerance.

## Figures and Tables

**Figure 1 ijerph-19-11864-f001:**
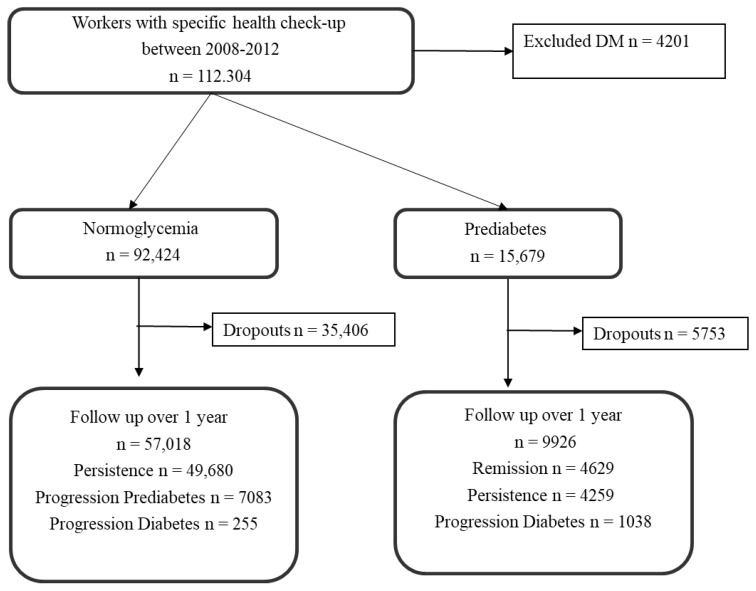
Flowchart of the study.

**Table 1 ijerph-19-11864-t001:** Comparison of baseline characteristics in individuals with normoglycemia and prediabetes.

	Normoglycemia		Prediabetes		*p*-Value
Number	57,018		9926		
Age (years)	36.0 ^†^	(30–45) ^‡^	49.0 ^†^	(39–57) ^‡^	<0.001
Male	30,427	(53.4)	6830	(68.8)	<0.001
BMI (kg/m^2^)	21.6 ^†^	(19.7–23.9) ^‡^	23.9 ^†^	(21.6–26.4) ^‡^	<0.001
Family history of diabetes	1329	(2.3)	249	(2.5)	0.281
Hypertension	17,048	(29.9)	5727	(57.7)	<0.001
Dyslipidemia	5195	(9.1)	3148	(31.7)	<0.001
Lifestyle behaviors					
Physical activities					
Less physically activity	34,132	(60.5)	6220	(63.6)	<0.001
Non-regular exercise	47,896	(84.5)	7884	(80.2)	<0.001
Not walking fast	29,706	(52.8)	5041	(52.0)	0.176
Eating speed					<0.001
Normal	33,188	(58.2)	5936	(59.8)	
Fast	17,853	(31.3)	3266	(32.9)	
Late dinner/snacking	26,691	(46.8)	4590	(46.2)	0.378
Skipping breakfast	17,249	(30.5)	2423	(24.7)	<0.001
Insufficient sleep	25,264	(44.8)	4179	(42.8)	<0.001
Smoking	18,328	(32.1)	3329	(33.5)	0.006
Heavy alcohol consumption	5348	(9.5)	1456	(14.9)	<0.001

Data are numbers (percentages) or median ^†^ (upper and lower quartiles) ^‡^. BMI, body mass index.

**Table 2 ijerph-19-11864-t002:** Association between lifestyle factors and change in glucose status during follow-up.

	Progress to Prediabetes from Normoglycemia during Follow-Up				Return to Normoglycemia from Prediabetes during Follow-Up			
	HR	95% CI	*p*-Value	*Z*-Value	HR	95% CI	*p*-Value	*Z*-Value
Baseline BMI (kg/m^2^)	1.08	1.07–1.09	<0.001	21.39	0.94	0.93–0.95	<0.001	−13.90
Change in BMI	1.05	1.03–1.07	<0.001	4.83	0.95	0.93–0.97	<0.001	−4.24
Less physical activity	0.98	0.93–1.03	0.396	−0.85	1.01	0.95–1.08	0.724	0.35
Non-regular exercise	1.05	0.98–1.12	0.171	1.37	0.99	0.91–1.07	0.724	−0.35
Not walking fast	1.01	0.96–1.06	0.716	0.36	0.97	0.91–1.03	0.270	−1.10
Eating speed (ref: “slow”)								
Normal	1.13	1.03–1.24	0.013	2.49	1.07	0.96–1.20	0.235	1.19
Fast	1.06	0.96–1.17	0.268	1.11	1.05	0.93–1.18	0.441	0.77
Late dinner/snacking	1.16	1.10–1.22	<0.001	5.80	0.98	0.92–1.04	0.430	−0.79
Skipping breakfast	1.12	1.06–1.18	<0.001	3.85	1.02	0.95–1.09	0.648	0.46
Insufficient sleep	1.01	0.96–1.06	0.721	0.36	0.99	0.93–1.06	0.820	−0.23
Smoking	0.98	0.93–1.04	0.564	-0.58	1.03	0.96–1.10	0.436	0.78
Heavy alcohol consumption	1.33	1.24–1.42	<0.001	8.19	1.05	0.96–1.14	0.265	1.11

HR, hazard ratio; CI, confidence interval; BMI, body mass index. Covariates were age, sex, family history of diabetes, hypertension, and dyslipidemia. Normoglycemia: Fasting plasma glucose < 5.6 mmol/L and HbA1c < 5.7%. Prediabetes: FPG ≥ 5.6–6.9 mmol/L and/or HbA1c ≥ 5.7–6.4%.

**Table 3 ijerph-19-11864-t003:** Sensitivity analysis of poor eating and exercise behaviors associated with progression to prediabetes from normoglycemia.

	HR	95% CI	*p*-Value	*Z*-Value	*p*-Trend
Baseline BMI (kg/m^2^)	1.08	1.07–1.08	<0.001	21.33	
Change in BMI	1.05	1.03–1.07	<0.001	5.16	
Poor eating behaviors (ref: “no”)					<0.001
Single	1.06	1.00–1.13	0.040	2.05	
Double	1.14	1.07–1.22	<0.001	3.88	
Triple	1.21	1.10–1.34	<0.001	3.80	
Poor exercise behaviors (ref: “no”)					0.308
Single	1.05	0.95–1.16	0.318	1.00	
Double	1.02	0.93–1.12	0.624	0.49	
Triple	1.06	0.97–1.16	0.218	1.23	

HR, hazard ratio; CI, confidence interval; BMI, body mass index. Covariates were age, sex, family history of diabetes, hypertension, dyslipidemia, insufficient sleep, smoking, and heavy alcohol consumption. Normoglycemia: Fasting plasma glucose < 5.6 mmol/L and HbA1c < 5.7%. Prediabetes: FPG ≥ 5.6–6.9 mmol/L and/or HbA1c ≥ 5.7–6.4%. Poor eating behaviors were defined as the sum of fast eating speed, late dinner/snacking, and skipping breakfast. Poor exercise behaviors were defined as the sum of less physical activity, not walking fast, and non-regular exercise.

## Data Availability

The datasets generated during and/or analyzed during the current study are available from the corresponding author upon reasonable request.

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
