# Peer review of "Poor Eating Behaviors Related to the Progression of Prediabetes in a Japanese Population: An Open Cohort Study"

_ijerph, 2022, doi:10.3390/ijerph191911864_

Round 1

Reviewer 1 Report

The study investigated the associations between some lifestyles, especially the eating behaviors, and the progression to prediabetes. The manuscript is generally well written. The organization is good and the sample size is large. All these lead the study valuable. In the following, I have listed some comments and reminders. Hope these are helpful.

1.     I understand that people might not take health check-up every year which caused a high proportion of dropouts. What is the influence on the study results? This should be noted.

2.     By excluding those with f/u < 1 y, all individuals included in the analysis were followed up over 1 year. However, the distribution of the f/u time was unclearly described.

3.     Eating behaviors are the key independent variables. The descriptions should be in more details. Only a simple sentence “Eating speed was categorized as fast, normal, and slow.” Is insufficient. The author could consider describing the question in the questionnaire, like the one of heavy alcohol consumption. My curious is whether it is perceived. This is related to the issue of validation of a measurement tool.

4.     Did any model diagnosis been made before the conclusion from Cox regression model?

5.     I was a little surprised that poor exercise behaviors were not associated with the change in glucose status (table 2). The explanation of such a result (in Discussion) seems having a room to be improved. Notice that the BMI and the change in BMI had been taking into account in the model.

6.     (Line 60) A baseline survey conducted between 2008 and 2012. The statement is inconsistent with it described in the flowchart.

7.     (Line 101) The definition of the change in BMI is unclear. Can you please explain it or modify the statement? (Not sure if it needs so many “means”?)

8.     (The footnote in Table 3) heavy alcohol “drinker” -> “consumption”. So that it is consistent with that given in Table 2.

Author Response

Response to Reviewer 1

We appreciate your thorough review of our manuscript. The comments have been helpful, and we have modified the manuscript based on the suggestions provided. Please find below our point-by-point responses to the reviewers’ comments.

Comment 1.     I understand that people might not take health check-up every year which caused a high proportion of dropouts. What is the influence on the study results? This should be noted.

Response: Thank you for your valuable comment. As you pointed out, we have noted in the Limitations that this issue might create a selection bias. (Please see the text marked in red on page 8, lines 276–279).

Comment 2.     By excluding those with f/u < 1 y, all individuals included in the analysis were followed up over 1 year. However, the distribution of the f/u time was unclearly described.

Response: Thank you for this comment. We have indicated the distribution of the f/u time in the Results section. (Please see the text marked in red on page 5, lines 155–157).

Comment 3.     Eating behaviors are the key independent variables. The descriptions should be in more details. Only a simple sentence “Eating speed was categorized as fast, normal, and slow.” Is insufficient. The author could consider describing the question in the questionnaire, like the one of heavy alcohol consumption. My curious is whether it is perceived. This is related to the issue of validation of a measurement tool.

Response: We appreciate the reviewer's comment regarding this issue. We have added more details on how the eating behaviors were defined by including the questions used in this study. (Please see the text marked in red on page 3, lines 88–95).

Comment 4.     Did any model diagnosis been made before the conclusion from Cox regression model?

Response: Thank you for this comment. We used the Schoenfeld residuals test and graphical methods to evaluate the Cox proportional hazard assumption. (Please see the text marked in red on page 4, lines 126–127).

Comment 5.     I was a little surprised that poor exercise behaviors were not associated with the change in glucose status (table 2). The explanation of such a result (in Discussion) seems having a room to be improved. Notice that the BMI and the change in BMI had been taking into account in the model.

Response: Thank you for your valuable comment. We have added more discussion on this point (Please see the text marked in red on page 7, lines 244–255).

Comment 6.     (Line 60) A baseline survey conducted between 2008 and 2012. The statement is inconsistent with it described in the flowchart.

Response: Thank you for pointing out this error. We have corrected Figure 1.

Comment 7.     (Line 101) The definition of the change in BMI is unclear. Can you please explain it or modify the statement? (Not sure if it needs so many “means”?)

Response: Thank you for pointing out this error. We have corrected the statement as follows: The change in BMI was defined as the value obtained by subtracting baseline BMI from BMI at onset of the outcome or the final follow-up (Please see the text marked in red on page 3, lines 102–104).

Comment 8.     (The footnote in Table 3) heavy alcohol “drinker” -> “consumption”. So that it is consistent with that given in Table 2.

Response: Thank you for pointing out this inconsistency. We have corrected this point in Table 3.

Thank you again for your feedback on our paper. We hope that the revised manuscript is now suitable for publication.

Reviewer 2 Report

The topic is extremely relevant for public health. Lifestyle factors that did not obviously lead to the progression of carbohydrate metabolism disorders were studied. This is indeed the first study examining disrupted eating behavior and alcohol use in relation to the progression of prediabetes in a Japanese population. Interesting and requiring more discussion are the data obtained in the present study that insufficient physical activity and smoking were not associated with the progression of prediabetes.

Notes and disadvantages:

It cannot be argued that normal values ​​of fasting glucose and glycated hemoglobin (HbA1c) determined once during dynamic observation are remission of prediabetes or diabetes. It is not known if multiple glucose and HbA1c controls were done to make sure there was indeed a remission. It is possible that no re-control was carried out. Still, in a journal with such a high quartile, the quality of scientific material should be better. This is certainly a good, important work, but it is rather weak for this journal.

The work would certainly have been better if the patients, albeit in a smaller sample, had undergone a continuous determination of glycated hemoglobin, glucose, and, if necessary, OGTT.

Here, one gets the feeling that the authors intentionally did not describe in detail the process of diagnosing carbohydrate metabolism disorders (CMD), because. simple observations were taken from the outpatient department, in which most of the patients were not subjected to sufficient diagnosis of CMD. In this case, we recommend either to significantly change the concept and terminology of the work and not to say that diabetes or prediabetes has regressed, and perhaps that diagnostic measures to establish diabetes and prediabetes were generally sufficient.

In this case, we can only say that when observed, the glucose values ​​​​were in the normal range, and in some cases, HbA1c was in the target ranges. But this does not mean that a remission of diabetes or prediabetes has occurred, there are completely different criteria for this. They are not described here, and most likely not observed. Only 25.6% of literary sources of the last 5 years.

I don’t know if the authors will be able to change the terms in such a way as to abandon the rather loud statement - “prediabetes remission”, “diabetes remission” in favor of more careful and correct ones. But then you will have to redo a significant part of the article, I'm not sure that this is possible.

Author Response

Responses to Reviewer #2

We wish to express our appreciation to the reviewer for their insightful comments, which have helped us significantly improve our manuscript.

Comment 1. It cannot be argued that normal values ​​of fasting glucose and glycated hemoglobin (HbA1c) determined once during dynamic observation are remission of prediabetes or diabetes. It is not known if multiple glucose and HbA1c controls were done to make sure there was indeed a remission. It is possible that no re-control was carried out. Still, in a journal with such a high quartile, the quality of scientific material should be better. This is certainly a good, important work, but it is rather weak for this journal.

The work would certainly have been better if the patients, albeit in a smaller sample, had undergone a continuous determination of glycated hemoglobin, glucose, and, if necessary, OGTT.

Response: Thank you for your comment. As you pointed out, it would be better to collect continuous data on glycated hemoglobin, glucose, or OGTT. However, the Japanese health checkup system only requires the measurement of blood glucose level or HbA1c as glucose metabolism index. Therefore, we do not have access to further information, such as continuous determination of glycated hemoglobin, glucose, and OGTT. We have acknowledged this shortcoming in the Limitations. (Please see the text marked in red on  page 8, lines 281–285).

Comment 2. Here, one gets the feeling that the authors intentionally did not describe in detail the process of diagnosing carbohydrate metabolism disorders (CMD), because. simple observations were taken from the outpatient department, in which most of the patients were not subjected to sufficient diagnosis of CMD. In this case, we recommend either to significantly change the concept and terminology of the work and not to say that diabetes or prediabetes has regressed, and perhaps that diagnostic measures to establish diabetes and prediabetes were generally sufficient.

In this case, we can only say that when observed, the glucose values ​​​​were in the normal range, and in some cases, HbA1c was in the target ranges. But this does not mean that a remission of diabetes or prediabetes has occurred, there are completely different criteria for this. They are not described here, and most likely not observed. I don’t know if the authors will be able to change the terms in such a way as to abandon the rather loud statement - “prediabetes remission”, “diabetes remission” in favor of more careful and correct ones. But then you will have to redo a significant part of the article, I'm not sure that this is possible.

Response: Thank you for your comment. As you pointed out, it is possible that misdiagnosis of prediabetes might have occurred in some cases because we relied on a single result of FPG or HbA1c test. Thus, we have acknowledged this shortcoming in the Limitations (Please see the text marked in red on page 8, lines 281–285).

 In addition, according to the reviewer`s comment, we have changed the terminology from “prediabetes remission” to “returned from prediabetes to normoglycemia” across the manuscript (We have marked this change in the manuscript with red text).

Comment 3. Only 25.6% of literary sources of the last 5 years.

Response: Thank you for your comment. We have cited more recent references (Please see the references indicated with red text).

Thank you again for your feedback on our paper. We hope that the revised manuscript is now suitable for publication.

Round 2

Reviewer 2 Report

The authors did a good job on the comments of the reviewers. The term "prediabetes regression" is no longer used in the text. The limitations of the study are described in more detail. The discussion is expanded, new interesting data are described based on recent publications, they are compared with their own results.

There remains a remark on statistics, it is important: it is possible to describe quantitative characteristics using the as mean ± standard deviation and consider differences as a t-test only if they have been tested for normal distribution. But very rarely the distribution in medical data turns out to be normal. In other cases, quantitative characteristics are described using the median and the upper and lower quartiles, and when comparing groups, non-parametric methods are used (the Mann Whitney test, for example). At a minimum, please add the name of the method used to check the normality of the distribution and the words that the distribution turned out to be normal. Otherwise, if the distribution of quantitative characteristics differs from the normal one, the statistical methods are applied incorrectly and, unfortunately, all the material will have to be recalculated.

Author Response

Response to the Reviewer’s comments

We appreciate your thorough review of our manuscript. The comments have been helpful, and we have modified the manuscript based on the suggestions provided. Please find below our point-by-point responses to the Reviewer’s comments.

Comment 1. There remains a remark on statistics, it is important: it is possible to describe quantitative characteristics using the as mean ± standard deviation and consider differences as a t-test only if they have been tested for normal distribution. But very rarely the distribution in medical data turns out to be normal. In other cases, quantitative characteristics are described using the median and the upper and lower quartiles, and when comparing groups, non-parametric methods are used (the Mann Whitney test, for example). At a minimum, please add the name of the method used to check the normality of the distribution and the words that the distribution turned out to be normal. Otherwise, if the distribution of quantitative characteristics differs from the normal one, the statistical methods are applied incorrectly and, unfortunately, all the material will have to be recalculated.

Response: Thank you for your comment. After checking the data distribution, we found that the age and BMI data were not normally distributed. Therefore, we have presented them as the median (the upper and lower quartiles) and analyzed them using the Mann-Whitney test (Please see the text marked in red on page 3, lines 115-117 and Table 1).

Thank you again for your feedback on our paper. We hope that the revised manuscript is now suitable for publication.